# Mitofusin 2-Deficiency Suppresses *Mycobacterium tuberculosis* Survival in Macrophages

**DOI:** 10.3390/cells8111355

**Published:** 2019-10-30

**Authors:** Junghwan Lee, Ji-Ae Choi, Soo-Na Cho, Sang-Hun Son, Chang-Hwa Song

**Affiliations:** 1Department of Medical Science, Chungnam National University, 266 Munhwa-ro, Jung-gu, Daejeon 35015, Korea; asrai1509@gmail.com (J.L.); jiae9035@gmail.com (J.-A.C.); wls3831@naver.com (S.-N.C.); yyhh666@naver.com (S.-H.S.); 2Department of Microbiology, College of Medicine, Chungnam National University, 266 Munhwa-ro, Jung-gu, Daejeon 35015, Korea; 3Research Institute for Medical Sciences, Chungnam National University, 266 Munhwa-ro, Jung-gu, Daejeon 35015, Korea

**Keywords:** MFN2, mitochondria, apoptosis, *Mycobacterium*, ER stress

## Abstract

Apoptosis is an important host defense mechanism against mycobacterial infection. However, the molecular mechanisms regulating apoptosis during mycobacterial infection are not well known. Recent reports suggest that bacterial infection regulates mitochondrial fusion and fission in various ways. Here, we investigated the role of mitochondria in *Mycobacterium tuberculosis* (Mtb)-infected macrophages. Mtb H37Rv (Rv) infection induced mitofusin 2 (MFN2) degradation, leading to mitochondrial fission. Interestingly, Mtb H37Ra (Ra) infection induced significantly greater mitochondrial fragmentation than Rv infection. Mtb-mediated Parkin, an E3 ubiquitin ligase, contributed to the degradation of MFN2. To evaluate the role of endoplasmic reticulum stress in the production of Parkin during Mtb infection, we analyzed Parkin production in 4-phenylbutyric acid (4-PBA)-pretreated macrophages. Pretreatment with 4-PBA reduced Parkin production in Mtb-infected macrophages. In contrast, the level of MFN2 production recovered to a level similar to that of the unstimulated control. In addition, Ra-infected macrophages had reduced mitochondrial membrane potential (MMP) compared to those infected with Rv. Interestingly, intracellular survival of mycobacteria was decreased in siMFN2-transfected macrophages; in contrast, overexpression of MFN2 in macrophages increased Mtb growth compared with the control.

## 1. Introduction

Tuberculosis (TB) is caused by *Mycobacterium tuberculosis* (Mtb), which can survive intracellularly [1]. Mtb is inhaled in aerosol form and infects lung alveolar macrophages [2]. Mtb-infected macrophages eliminate Mtb by inducing apoptosis [2]. Many intracellular bacteria use host organelles such as the nucleus, mitochondria, Golgi, and endoplasmic reticulum (ER) [3]. Mtb changes the shape of mitochondria, leading to mitochondrial dysfunction [4]. Mitochondria are highly dynamic organelles that build the large interconnected intracellular networks responsible for cellular metabolism, differentiation, signaling, and death [5]. Mitochondrial dynamics are controlled by their fusion and fission [5]. Mitochondrial fusion involves the outer mitochondrial membrane GTPases mitofusin 1/2 (MFN1/2), and inner membrane GTPase optic atrophy 1 (OPA1) [5]. Mitochondrial fission requires mitochondrial fission 1 protein (FIS1) and GTPase dynamin-related protein 1 (DRP1), but the molecular details of fusion and fission are unknown [5]. Mitochondrial fission seems to be an early stage of apoptosis [6,7,8,9]. Additionally, mitochondrial dynamics function as signals to induce the innate immune response during viral infection [10,11]. However, the role of mitochondrial dynamics during infection is unclear.

Intracellular pathogens regulate cell death by modulating host mitochondria. Human hepatitis B virus (HBV)-induced fission through DRP1 increases mitophagy, promoting cell survival and HBV replication [12]. Dengue virus (DENV) infection also destroys mitochondrial dynamics by inducing DRP1-dependent fission, which supports DENV replication [13]. However, some bacteria suppress fission to promote their intracellular survival. The knockdown of DRP1 increases intracellular survival of bacteria; conversely, knockdown of MFN1/2 decreases intracellular survival during *Listeria monocytogenes* infection [14]. In addition, avirulent Mtb H37Ra (Ra) induces more severe mitochondrial dysfunction than virulent Mtb H37Rv (Rv), and Mtb-induced ultrastructural changes in mitochondria [15]. However, the relationship between Mtb and mitochondrial dynamics in host cells is unclear.

In mammalian cells, the E3 ubiquitin ligase Parkin regulates mitochondrial dynamics by promoting proteasome-dependent degradation of MFN1/2 [16]. Parkin is involved in the innate immune response to *Mycobacterium leprae* and Mtb infection by promoting ubiquitin-mediated autophagy of mycobacteria and inhibiting mycobacterial replication in macrophages [17,18].

ER stress is implicated in the response to Mtb [19,20,21] and regulates apoptosis in the presence of a variety of pathogens [22,23,24]. However, the relationship between ER stress-mediated apoptosis and mitochondrial dysfunction is unclear. Here we report that the interplay of ER stress and mitochondrial dynamics promotes apoptosis during mycobacterial infection of murine macrophages.

## 2. Materials and Methods

### 2.1. Cell Culture

Primary bone marrow-derived macrophages (BMDMs) from C57BL/6 mice and murine macrophage RAW 264.7 cells (American Type Culture Collection; ATCC) were isolated and cultured as described previously [21]. BMDMs were isolated and differentiated by culturing for five days in medium containing 2 ng/mL macrophage colony-stimulating factor (R&D Systems, Minneapolis, MN, USA). Before infection the cells were cultured in 12- or 24-well polypropylene tissue culture plates in 5% CO_2_ at 37 °C for 24 h to enable adherence. All animal procedures were reviewed and approved by the institutional animal care and use committee of Chungnam National University, Daejeon, Korea (permit no. CNU-00425). The animal experiments were performed in accordance with the Korean Food and Drug Administration guidelines.

### 2.2. Mtb Culture and Infection

Mtb H37Rv (ATCC), H37Ra (ATCC), Rv-RFP, and Ra-RFP were cultured and infected as described [21]. Mtb was grown in Middlebrook 7H9 medium (BD Biosciences, Franklin Lakes, NJ, USA) supplemented with 10% OADC and 5% glycerol. Rv- and Ra-RFP were cultured in Middlebrook 7H9 medium supplemented with 10% OADC and selected using 50 μg/mL kanamycin (Sigma-Aldrich, St. Louis, MO, USA). The Mtb strains were stored at −80 °C until use. Cells were infected with Mtb at a multiplicity of infection (MOI) of 1 and incubated for 3 h at 37 °C in 5% CO_2_. After allowing time for phagocytosis, the cells were washed with PBS to remove extracellular bacteria and incubated with fresh medium without antibiotics. To assay intracellular survival, BMDMs were infected with Mtb for 24 or 48 h and lysed in distilled water to collect intracellular bacteria. The lysates were plated separately on 7H10 agar (BD Biosciences) plates and incubated for three weeks at 37 °C.

### 2.3. Western Blotting Analysis

Western blotting was performed as described previously [20]. Whole-cell lysates were separated by 12% sodium dodecyl sulfate-polyacrylamide gel electrophoresis (SDS-PAGE) and transferred to a polyvinylidene difluoride (PVDF) membrane. After incubation with the appropriate antibodies, bound antibodies were detected using chemiluminescent HRP substrate (Millipore, St. Louis, MO, USA). The blots were exposed and quantified using an Alliance Mini 4M (UVITEC Cambridge, Cambridge, UK). The primary antibodies were as follows: Anti-MFN1/2, anti-OPA1, anti-DRP1, anti-MFF, anti-GRP78/BiP, anti-CHOP, anti-phospho-eIF2a, anti-caspase-3 (Cell Signaling Technology, Danvers, MA, USA), and anti-β-actin (Santa Cruz Biotechnology, Dallas, TX, USA). The secondary antibodies were anti-rabbit IgG-HRP (Cell Signaling Technology) and anti-mouse-IgG-HRP (Calbiochem, Darmstadt, Germany).

### 2.4. Reagents

An ATF6 inhibitor (AEBSF; Sigma-Aldrich), IRE1 inhibitor (Irestatin; Axon, Groningen, Netherlands), PERK inhibitor (GSK2606414; Millipore), proteasome inhibitor (MG-132; Sigma-Aldrich), 4-phenylbutrate (4-PBA; Sigma-Aldrich), and tunicamycin (Tm; Millipore) were dissolved in DMSO (Sigma-Aldrich) and diluted to the desired concentration in culture medium. Macrophages were pretreated with the indicated concentrations of inhibitors or inducers for 1 h prior to Mtb infection.

### 2.5. Transfection of Small Interfering RNA

Silencing of MFN2 and Parkin was achieved using small interfering RNAs (siRNAs) (200 nM) for mouse MFN2 mRNA target sequences (Santa Cruz Biotechnology), mouse Parkin mRNA target sequences (Santa Cruz Biotechnology), and negative control siRNAs (Bioneer, Daejeon, Korea). The siRNA oligonucleotides were transfected into cultured BMDMs using Lipofectamine 3000 (Invitrogen, Carlsbad, CA, USA) according to the manufacturer’s instructions.

### 2.6. Immunofluorescence

BMDMs were grown on 18-mm coverslips for 24 h at 37 °C. After Mtb infection for 3 h, the cells were fixed in 4% paraformaldehyde and washed three times with TBS-T. Next, the cells were blocked in 5% skim milk for 1 h at room temperature. The cells were cultured with primary antibodies overnight, and then with the appropriate secondary antibody (Alexa Fluor 594 anti-rabbit IgG, Alexa Fluor 594 anti-mouse IgG, Alexa Fluor 488 anti-rabbit IgG, and Alexa Fluor 488 anti-mouse IgG, Life Technologies, Carlsbad, CA, USA) for 2 h at room temperature. Next, the cells were stained with DAPI to label DNA. The stained cells were visualized under a DP70 fluorescence microscope (400× magnification; Zeiss, Oberkochen, Germany). Quantification of mitochondrial morphology was measured using ImageJ (NIH) as described previously [25,26].

### 2.7. MFN2 Cloning

Primers for MFN2 cloning were manufactured by Cosmo Genetech (Seoul, Korea). For ligation with the vector, amplified PCR products were incubated for 150 s at room temperature and 10 min on ice. After ligation, transformation was performed with *Escherichia coli* Top10 cells for 20 min on ice, 90 s at 42 °C, and 10 min on ice. The products were cultured in LB medium for 1 h at 37 °C and centrifuged for 1 min at 13,000 rpm. After removal of the supernatant, resuspended cells were incubated for 24 h at 37 °C in LB agar containing ampicillin. Selected colonies were cultured in LB medium containing ampicillin for 24 h at 37 °C. The plasmids were isolated using a Plasmid DNA Mini-Prep Kit (ELPIS Biotech, Daejeon, Korea) in accordance with the manufacturer’s protocol. Sequencing was performed by Cosmo Genetech. BMDMs were transfected with a mouse MFN2 expression vector (pcDNA3.1-MFN-2-V5) and empty vector (pcDNA3.1) using Lipofectamine 3000 (Invitrogen), and incubated with fresh complete medium containing 5% FBS without antibiotics.

### 2.8. Apoptosis Assay

Cell death was assessed using an Annexin V/propidium iodide (PI) staining kit according to the manufacturer’s instructions (BD Biosciences). The cells were stained with FITC-conjugated Annexin V and propidium iodide (PI) and subjected to flow cytometry using a FACSCanto II instrument with FACS Diva, and the results were analyzed using FlowJo software (BD Biosciences).

### 2.9. Oxygen Consumption Rate Assay

BMDMs were plated in XF24 cell culture plates. The oxygen consumption rate (OCR) was measured using XF Running Buffer and an XF24 Extracellular Flux Analyzer (Agilent Technologies, Santa Clara, CA, USA). Oligomycin (1 μM) was injected through port A, followed by CCCP (5 μM) through port B, and rotenone + antimycin A, both at a final concentration of 0.5 μM (all from Agilent Technologies), via port C.

### 2.10. Statistical Analyses

Each experiment was performed at least three times. Statistical significance was analyzed using GraphPad Prism 5 software by one-way analysis of variance (ANOVA), Friedman test, or Wilcoxon signed-rank test. Statistical significance was indicated by * *p* < 0.05, ** *p* < 0.01, and *** *p* < 0.001.

## 3. Results

### 3.1. Mtb Infection Modulates Mitochondrial Function

To examine the functional changes in mitochondria during Mtb infection, we used JC-1 dye to assay mitochondrial membrane potential (MMP) by determining the ratio of the aggregated (red fluorescence) to monomeric (green fluorescence) form. The amount of mitochondrial aggregates was decreased in Mtb-infected BMDMs at 48 h (Figure 1A). The red/green ratio of Mtb-infected BMDMs was reduced compared to that of the unstimulated control (Figure 1B). Infection with Ra induced greater green fluorescence than infection with Rv (Figure 1A,B). The ratio of phagocytosis between Ra and Rv was not different (Appendix A). Next, we analyzed the MMP of BMDMs infected with Rv or Ra by flow cytometry. The MMP was significantly decreased in Rv-infected BMDMs and Ra-infected BMDMs compared to unstimulated control cells (Figure 1C). To investigate the effects of changes in mitochondrial morphology on mitochondrial respiration during Mtb infection, we measured the mitochondrial oxygen consumption rate (OCR) of Mtb-infected BMDMs at 48 h post-infection. The basal OCR was reduced in Mtb-infected BMDMs compared with that in the unstimulated control and was significantly lower in Ra- than Rv-infected macrophages (Figure 1D). At 48 h, mitochondrial ATP production was decreased in Mtb-infected BMDMs compared to the unstimulated control and was lower in Ra- than Rv-infected BMDMs (Figure 1E). These results suggest that mitochondrial function is altered during Mtb infection of BMDMs.

To investigate whether Mtb modulates mitochondrial fragmentation in macrophages, we analyzed the mitochondrial morphology of Mtb-infected BMDMs by confocal microscopy at 48 h post-infection. A fragmented mitochondrial morphology was observed in Rv- and Ra-infected cells compared to control cells (Figure 2A). Rv infection induced greater mitochondrial fragmentation than Ra infection (Figure 2A). Next, we measured the levels of mitochondrial fission and fusion proteins during Mtb infection. The level of MFN2 was decreased in mycobacteria-infected macrophages at 48 h. However, the levels of the mitochondrial fusion proteins MFN1 and OPA were not significantly reduced in Mtb-infected macrophages. In contrast, the levels of the mitochondrial fission proteins DRP1 and p-MFF were increased in Mtb-infected macrophages. Western blotting showed that mitochondrial fission was induced in Mtb-infected cells compared with the control (Figure 2C,D). In addition, we found that production of MFN2 was decreased in a dose-dependent manner during Mtb infection (Figure 2E). These findings suggest that Mtb affects the production of mitochondrial fusion and fission proteins in macrophages.

### 3.2. MFN2 is Sensitive to the Survival of Attenuated Ra

Parkin, an E3 ubiquitin ligase, eliminates damaged mitochondria via the ubiquitin-mediated proteasome pathway and plays a role in removing Mtb from macrophages [18]. We investigated whether Parkin is involved in MFN2 production in Mtb-infected BMDMs. MFN2 production was significantly decreased in Mtb-infected macrophages at 48 h compared to the control. In contrast, Parkin production was increased in Mtb-infected macrophages at 48 h (Figure 3A,B). Interestingly, Ra-induced Parkin production was greater than that induced by Rv. Next, we examined MFN2 production after RFP-Mtb infection using confocal microscopy. As expected, MFN2 levels were decreased in Mtb-infected macrophages compared to the control (Figure 3C). Ra-induced MFN2 production was more reduced than that induced by Rv. To investigate whether Parkin is involved in Mtb-induced production of MFN2, we assayed Mtb-induced MFN2 production in the presence of Parkin siRNA. MFN2 production was restored in Rv- and Ra-infected macrophages compared to the control (Figure 3D,E). In addition, increased levels of mitochondrial fragmentation by Mtb infection were reduced in siParkin-transfected BMDMs (Figure 3F and Appendix A). Mtb-induced Parkin regulated MFN2 production, indicating that Parkin plays a role in the degradation of MFN2 via the ubiquitin-mediated proteasome pathway.

Because ER stress upregulates Parkin production in Parkinson disease [27], we examined the relationship between ER stress and Parkin during Mtb infection. Ra-infected macrophages showed increased production of ER stress sensor molecules such as CHOP, p-eIF2α, and BiP in a time-dependent manner compared to those infected with Rv (Figure 4A,B). Mtb-induced Parkin production was strongly induced in Tm-pretreated macrophages (Figure 4C). Therefore, ER stress affects Parkin production during Mtb infection, leading to MFN2 degradation. Moreover, Mtb-induced Parkin production was reduced by pretreatment with 4-PBA (a chemical chaperone), suggesting that ER stress-mediated production of Parkin plays an important role in the regulation of MFN2 production during Mtb infection (Figure 4D). To identify the ER stress pathway (protein kinase R-like ER kinase [PERK], inositol-requiring kinase 1 [IRE1], and activating transcription factor 6 [ATF6]) involved in inducing Parkin production in Mtb-infected macrophages, we analyzed Parkin production in the presence of GSK2606414 (PERK inhibitor), IREstatin (IRE1 inhibitor), and AEBSF (ATF6 inhibitor). Interestingly, all of the inhibitors reduced Rv- and Ra-induced Parkin production (Figure 4E,F). These results suggest that Mtb regulates MFN2 production in macrophages via the ER stress pathway.

Parkin ubiquitinates MFN2, leading to its proteasomal degradation [28]. To investigate whether MFN2 degradation is dependent on MFN2 ubiquitination during Mtb infection, we measured ubiquitin (Ub) production in Mtb-infected macrophages by western blotting. Mtb induced production of Ub at 24 and 48 h (Figure 5A). We hypothesized that if Mtb-induced ubiquitination is associated with ER stress, the ER stress pathway may be required for Parkin-induced ubiquitination. Next, we determined which ER stress pathway is associated with Mtb-induced ubiquitination. As expected, Mtb-induced Ub production was reduced by all of the ER-stress inhibitors (Figure 5B). We next examined MFN2 production in Mtb-infected macrophages treated with MG132, a proteasome inhibitor. MG132 prevented MFN2 degradation during Mtb infection (Figure 5C,D). These results indicate that Parkin regulates MFN2 production via the proteasome-Ub system in Mtb-infected macrophages.

### 3.3. Effect of MFN2 on Intracellular Survival of Mycobacteria

To investigate whether MFN2 production affects intracellular survival of Mtb in macrophages, we silenced or overexpressed MFN2 in macrophages. We examined apoptosis after infection with Rv or Ra in MFN2-silenced macrophages using Annexin V/PI staining. The proportion of apoptotic Rv-infected siMFN2-silenced macrophages (24.9% ± 3.2%) was significantly higher than that of the control (16.9% ± 2.1%). Similarly, apoptosis was significantly increased in Ra-infected siMFN2-silenced macrophages (46.2% ± 1.9%) compared to the control (39.7% ± 2.0%) (Figure 6A). The number of CFU was significantly reduced in Rv-infected siMFN2-transfected macrophages compared to the control (Figure 6B). Similarly, the number of CFU was lower in Ra-infected siMFN2-silenced macrophages than in the control (Figure 6C). In contrast, Rv-infected MFN2-overexpressing macrophages exhibited a lower frequency of apoptosis (9.1% ± 2.4%) compared to the control (16.5% ± 6.5%). The frequency of apoptosis of Ra-infected MFN2-overexpressing macrophages (30.9% ± 4.7%) was decreased compared to the control (36.8% ± 6.9%) (Figure 6D). Interestingly, survival of Rv and Ra was significantly increased in MFN2-overexpressing macrophages compared to the control (Figure 6E,F). Taken together, these data suggest that MFN2 modulates the intracellular survival of Mtb by regulating apoptosis.

## 4. Discussion

Several bacterial taxa, such as enteropathogenic *Escherichia coli*, *Helicobacter pylori*, *Staphylococcus aureus*, and *Shigella flexneri*, survive by preventing apoptosis [29]. Disruption of the mitochondrial fusion/fission balance leads to increased mitochondrial fragmentation [30,31]. In this study, we showed that infection with attenuated Ra Mtb induced greater mitochondrial fragmentation than infection with Rv Mtb. However, the levels of MMP, OCR, and ATP were significantly lower in Ra-infected macrophages than in those infected with Rv. Mitochondrial dysfunction is reportedly increased in Ra- compared to Rv-infected THP-1 cells, decreasing the MMP and ATP synthesis [15]. Ra-induced mitochondrial fragmentation may be due to disruption of the mitochondrial fusion/fission balance in macrophages. Therefore, lower levels of MMP, OCR, and ATP may be associated with mitochondrial fragmentation during Mtb infection.

Production of MFN2 was significantly reduced in Ra- compared with Rv-infected macrophages. In contrast, the levels of the fission proteins DRP1 and MFF were increased in Ra-infected macrophages compared to the control. DRP1 induces cytochrome *c* release during apoptosis and MFN1/2 delays Bax activation, cytochrome *c* release, and apoptosis [32]. The difference in mitochondrial fragmentation between infection with Rv and Ra is likely dependent on the levels of fusion/fission protein production. Therefore, our data suggest that increased mitochondrial fragmentation may be caused by reduced MFN2 production during Mtb infection.

Parkin (an E3 ligase) reduces MFN2 production by ubiquitination in damaged mitochondria [33]. A recent study showed that Parkin induction limited Mtb survival through autophagy of Mtb-infected macrophages [18]. However, the relationship between Parkin and MFN2 is unclear during Mtb infections. In this study, Mtb-mediated Parkin production decreased MFN2 production via ubiquitination. The production of Parkin is induced by ER stress [34]. Although ER stress plays an important role in mycobacterial infection, Ra-mediated ER stress was stronger than that induced by the virulent strain Rv [19,20,21]. Also, Ra induced a higher level of Parkin production than Rv, possibly due to different intensities of ER stress. Compared to Rv, ER stress response is strong in Ra-infected macrophages. It sequentially induced Parkin production and eventually resulted in strong apoptosis. Three ER stress sensor molecules were involved in Parkin production: PERK, IRE1, and ATF6. These contribute to the generation of reactive oxygen species [35,36], which triggers mitochondrial recruitment of Parkin [16,37]. Our data suggest that ER stress-mediated Parkin is a MFN2 regulator during mycobacterial infection in macrophages.

Knockdown of MFN2 induces apoptosis by disrupting the mitochondrial network, suppressing the growth of Mtb. A low level of MFN2 decreases the MMP, leading to mitochondrial dysfunction and apoptosis [38,39]. Therefore, our data suggest that regulation of MFN2 levels is important for suppressing the survival of Mtb in macrophages.

In summary, we found that Mtb-induced ER stress increases mitochondrial fragmentation and apoptosis. MFN2 plays a key role in disrupting mitochondria, which suppresses the growth of Mtb in macrophages. Therefore, MFN2 may be a target for host-derived therapeutics against mycobacterial infection.

## Figures and Tables

**Figure 1 cells-08-01355-f001:**
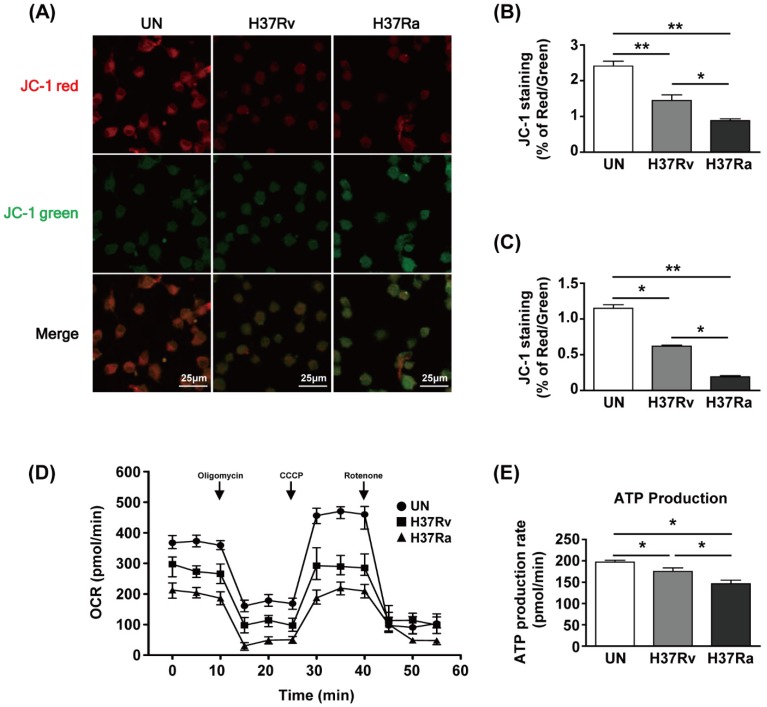
*Mycobacterium tuberculosis* (Mtb) increases mitochondrial dysfunction in macrophages. Bone marrow-derived macrophages (BMDMs) were infected with Mtb H37Rv (Rv) or Mtb H37Ra (Ra) (MOI = 1) for 3 h and incubated for 24 and 48 h. Mitochondrial membrane potential in BMDMs was assessed by JC-1 staining. Fluorescence of JC-1 was detected at an excitation wavelength of 488 nm and an emission wavelength of 530 nm by (**A**) confocal microscopy or (**C**) flow cytometry. (**B**) Quantification of the red-to-green ratio in (**A**). (**D**) Oxygen consumption rate (OCR) after sequential treatment with oligomycin, carbonyl cyanide 3-chlorophenylhydrazone (CCCP), and rotenone. (**E**) ATP production in Mtb-infected BMDMs. ATP production rate was calculated from OCR measured in the XF24 Extracellular Flux Analyzer. ATP production rate was calculated from OCR measured in the XF24 Extracellular Flux Analyzer. Data are means ± SD of three independent experiments. * *p* < 0.05, ** *p* < 0.01.

**Figure 2 cells-08-01355-f002:**
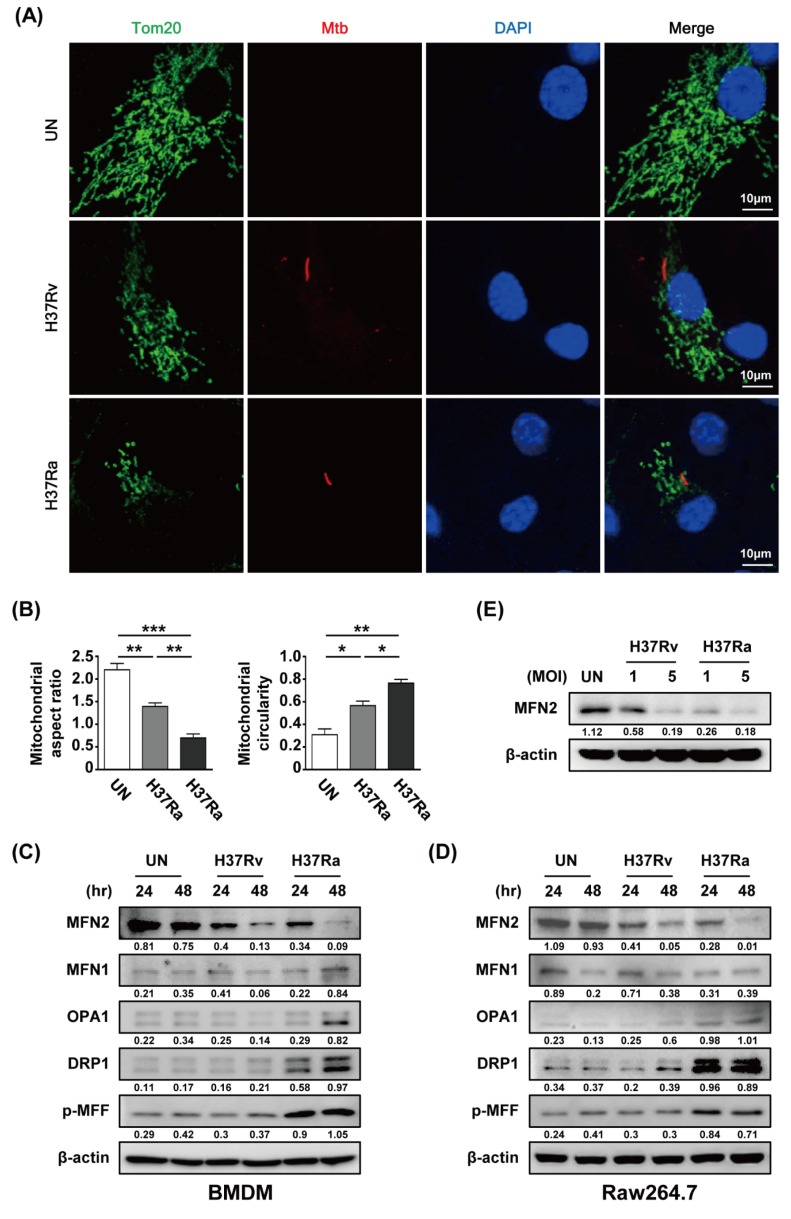
Mtb induces mitochondrial fragmentation in macrophages. (**A**) BMDMs were infected with RFP-labeled Rv or Ra (red), incubated for 48 h, stained with Tom20 to detect mitochondria (green) and DAPI to visualize nuclei (blue), and visualized using confocal microscopy. (**B**) Quantification of the mitochondrial morphology (aspect ratio; left, and circularity; right) in (**A**). (**C**,**D**) Cells were harvested and subjected to western blotting for mitochondrial fusion/fission factors (such as MFN1/2, OPA1, Drp1, and p-MFF); β-actin was used as the loading control. (**E**) BMDMs were infected with Rv or Ra (MOI = 1 to 5) for 3 h, and then incubated 48 h. Western blotting analysis was performed using antibodies directed against MFN2 and β-actin. Data are representative of three independent experiments. Statistically significant differences are indicated; * *p* < 0.05, ** *p* < 0.01, and *** *p* < 0.001.

**Figure 3 cells-08-01355-f003:**
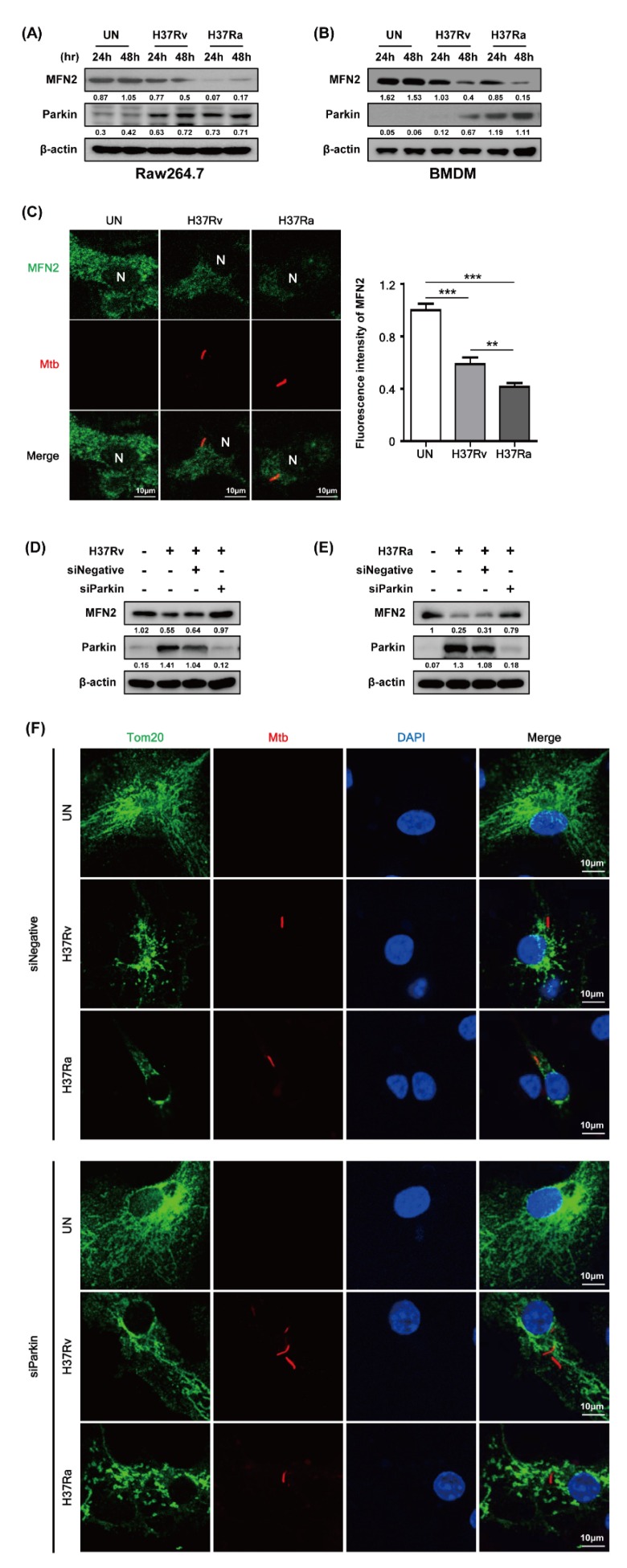
Mtb infection decreases mitofusin 2 (MFN2) production via Parkin induction. (**A**,**B**) Raw264.7 cells or BMDMs were infected with Rv or Ra (MOI = 1) for 3 h. After washing, the cells were incubated for 24 and 48 h. MFN2 and Parkin were detected by western blotting. (**C**) Representative confocal images of Mtb-infected cells. DNA was stained with DAPI (blue) and MFN2 (green). BMDMs were transfected with siNegative (200 nM) or siParkin (200 nM), and infected with Rv or Ra (MOI = 1; 48 h). In siRNA-transfected BMDMs, we measured the (**D**,**E**) expression of MFN2 and Parkin using western blotting, and (**F**) mitochondrial fragmentation by confocal microscopy. Data are representative of three independent experiments. Statistically significant differences are indicated; ** *p* < 0.01, and *** *p* < 0.001.

**Figure 4 cells-08-01355-f004:**
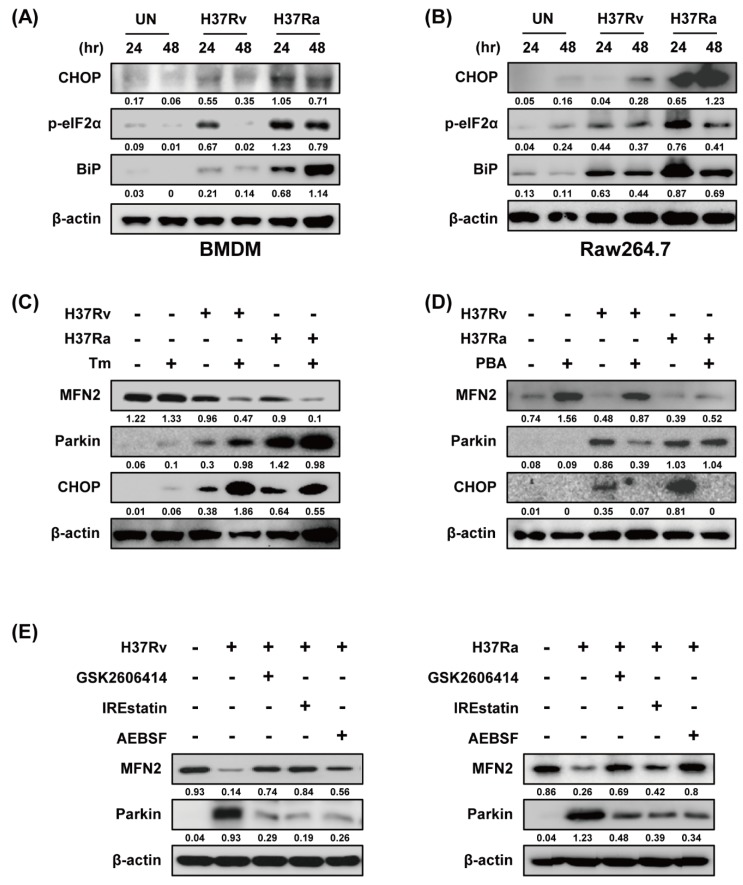
Mtb-induced endoplasmic reticulum (ER) stress modulates MFN2 production in macrophages. (**A**,**B**) Raw264.7 cells and BMDMs were infected with Mtb at a MOI of 1 for 3 h and CHOP, p-eIF2α, and BiP were detected by western blotting. β-actin was used as the loading control. BMDMs were pretreated with (**C**) tunicamycin (Tm, 500 ng/mL), (**D**) 4-phenylbutyric acid (4-PBA, 10 mM), or (**E**) ER stress inhibitors (10 µM GSK2606414, 5 µM IREstatin, 500 µM AEBSF) for 1 h and infected with Mtb for 3 h. At 48 h after Mtb infection, western blotting for Parkin, MFN2, CHOP, and β-actin was performed. Data are representative of three independent experiments.

**Figure 5 cells-08-01355-f005:**
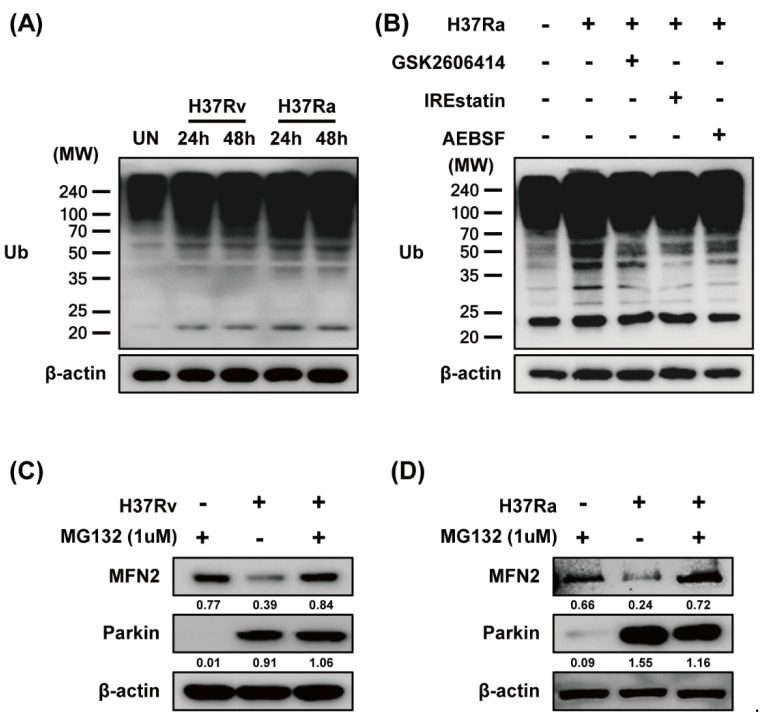
Mtb-induced Parkin regulates MFN2 production via the ubiquitin-proteasome system. (**A**) BMDMs were infected with Rv or Ra for 3 h. The cell lysates were subjected to western blotting for ubiquitin. (**B**) BMDMs in the presence or absence of ER stress inhibitors (10 µM GSK2606414, 5 µM IREstatin, 500 µM AEBSF) or (**C**,**D**) MG132 for 1 h, before infection with Rv or Ra. At 48 h after Mtb infection, western blotting for ubiquitin, Parkin, and MFN2 was performed. β-actin was used as the loading control. Data are representative of three independent experiments.

**Figure 6 cells-08-01355-f006:**
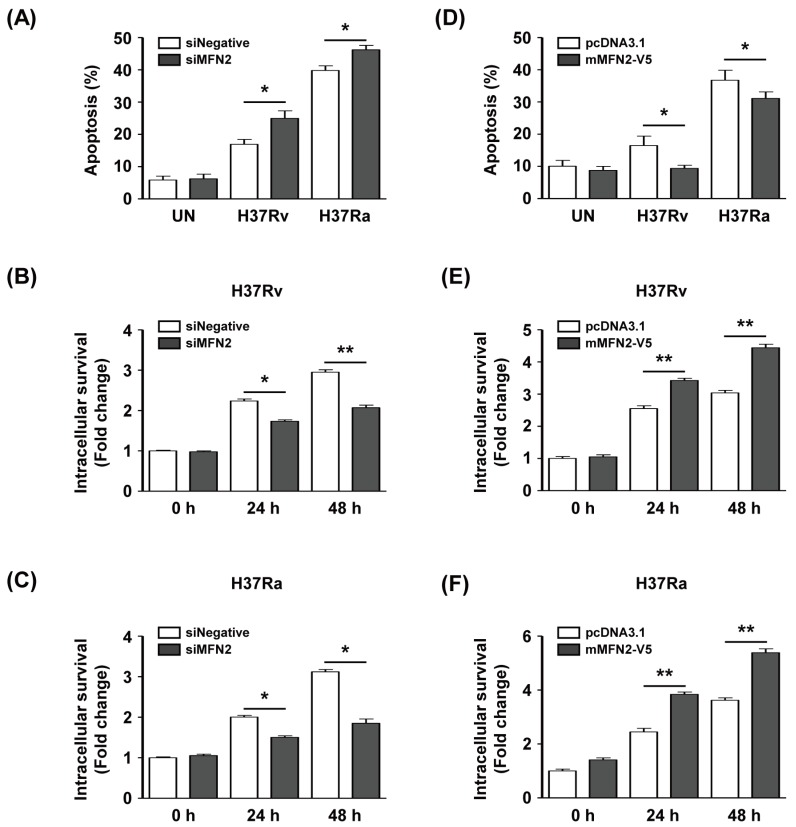
MFN2 regulates intracellular survival of Mtb. BMDMs were transfected with siRNA (siNegative or siMFN2, 200 nM). (**A**) Apoptosis and (**B**,**C**) intracellular survival of siRNA-treated BMDMs infected with Rv or Ra for 24 and 48 h. (**D**) BMDMs were transfected with pcDNA3.1-MFN2, infected with Mtb (MOI = 1) for 24 and 48 h, and apoptosis was analyzed by flow cytometry. (**E**,**F**) MFN2-overexpressing cells were infected with Rv or Ra (MOI = 1) for 24 and 48 h, and intracellular survival was assayed by enumerating CFU. Intracellular survival of Mtb was expressed as the change (n-fold) in the bacterial number at a given time point relative to the initial number of invasive bacteria. The sum of Annexin V^+^/PI^−^ and Annexin V^+^/PI^+^ cells were considered apoptotic. Data are means ± SD of three independent experiments. * *p* < 0.05, ** *p* < 0.01.

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
