# Peer review of "Mitofusin 2-Deficiency Suppresses Mycobacterium tuberculosis Survival in Macrophages"

_cells, 2019, doi:10.3390/cells8111355_

Round 1

Reviewer 1 Report

This paper by Lee et al deals with the role of apoptosis and M. tuberculosis survival in murine macrophages. The overall approach is sound but the final conclusion that MFN2 could be a drug target for Mtb treatment is somewhat far reached. Since modulation of MFN2 and Parkin have only moderate effects on apoptosis and intracellular survival, other important mechanisms are involved.

Specific comments:

It is well known that attenuated Mtb Ra induces apoptosis and does not survive intracellularly, in contrast to Mtb Rv. However, to analyze how Ra/Rv trigger ER stress and mitochondrial fragmentation, it is vital to know if the infectious rate is the same.  Analysing oxygen consumption (fig 1D/E), OCR pm/min is used both for oxygen consumption and ATP. Is that correct? In fig 2A mitochondrial fragmentation is visualized with confocal microscopy. It is stated that there is a clear difference in mitochondrial fragmentation between Rv and Ra. This is not very convincing from the images. When analyzing MFN2, there is a discrepancy between the data in fig 2B and 3B.  The siRNA treatment clearly increases MFN2 . It would be valuable to show if the treatment affected mitochondrial fragmentation as well. It is stated that overexpression of MFN2 in RV-infected cells increased apoptosis - should be decreased (16.5 to 9.1%). 

Author Response

Reviewer 1.

It is well known that attenuated Mtb Ra induces apoptosis and does not survive intracellularly, in contrast to Mtb Rv. However, to analyze how Ra/Rv trigger ER stress and mitochondrial fragmentation, it is vital to know if the infectious rate is the same. 

The BMDMs were infected with MtbH37Rv or MtbH37Ra for 3h at MOI of 1:1 or 1:5. Then, BMDMs were washed to remove extracellular bacteria and lysed in distilled water to collect intracellular bacteria. The number of Mtb was determined by CFU counting. In this study, phagocytosis ratio of Mtb was about 20% in macrophages. There was no different phagocytosis ratio between Ra and Rv. We revised supplementary figure 1. We added the text as follows;

“The ratio of phagocytosis between Ra and Rv was not different (Supplementary figure 1).”

Analysing oxygen consumption (fig 1D/E), OCR pm/min is used both for oxygen consumption and ATP. Is that correct?

OCR was measured in an XF24 Extracellular Flux Analyzer and then ATP production was calculated from OCR. To be clarified, we revised “OCR pm/min” to “ATP production rate” in figure 1E. We added the text as follows in figure legend;

ATP production rate was calculated from OCR measured in the XF24 Extracellular Flux Analyzer.”

In fig 2A mitochondrial fragmentation is visualized with confocal microscopy. It is stated that there is a clear difference in mitochondrial fragmentation between Rv and Ra. This is not very convincing from the images.

To be clarified, mitochondrial fragmentation was quantified and shown in fig 2B. The mitochondrial aspect ratio and circularity were measured using ImageJ. We added the text as follows in figure 2 legend and materials and methods;

“(B) Quantification of the mitochondrial morphology (aspect ratio; left, and circularity; right) in (A).”

“Quantification of mitochondrial morphology was measured using ImageJ (NIH) as described previously [25,26].”

When analyzing MFN2, there is a discrepancy between the data in fig 2B and 3B.

We revised the figures with good quality in figure 2C.

The siRNA treatment clearly increases MFN2. It would be valuable to show if the treatment affected mitochondrial fragmentation as well.

Thank you for your advice. We show that mitochondrial fragmentation by both Rv and Ra were reduced in Parkin knockdown macrophages (Fig 3F). We added the text in results as fllows;

“In addition, increased mitochondrial fragmentation by Mtb infection was reduced in siParkin-transfected BMDMs (Figure 3F and supplementary figure 2).”

It is stated that overexpression of MFN2 in RV-infected cells increased apoptosis - should be decreased (16.5 to 9.1%). 

It was our mistake. We revised the text as follows;

“In contrast, Rv-infected MFN2-overexpressing macrophages exhibited a lower frequency of apoptosis compared to the control.”

Reviewer 2 Report

The title of the manuscript is misleading. As per the results, MFN2 is heavily reduced by Mtb H37Ra and not by H37Rv. Although H37Ra is Mtb, it is an attenuated strain. This should be, somehow, conveyed in the title. For infecting cells, only one MOI is constantly used in all studies. To make sure that decrease/increase in proteins are due to infection, the authors should have used different MOI of infection at least in one place, may be at the beginning.. The immunoblot results are not convincing. The B-actin levels vary from lane to lane in all blots. May be the authors should quantitate the levels of proteins with Image J or with similar software and express the reduction/increase in relation to B-actin. This is applicable to all immunoblots. Otherwise, the blot results are not valid. In section 3.2, the sub-title needs to be modified as "Mtb-mediated MFN2 decrease is associated". The manuscript does not discuss why the attenuated H37Ra strain is more effective in affecting the mitochondrial proteins than the virulent H37Rv strain. This is a major drawback.

Author Response

Reviewer 2

The title of the manuscript is misleading. As per the results, MFN2 is heavily reduced by Mtb H37Ra and not by H37Rv. Although H37Ra is Mtb, it is an attenuated strain. This should be, somehow, conveyed in the title.

In Fig. 6, our results showed that MFN2 deficiency suppressed both Mtb H37Rv and Mtb H37Ra survival in macrophages. We agree with your opinion that attenuated Mtb strain is more sensitive to MFN2 expression. Therefore, we toned down the title with “Mitofusin 2-deficiency suppresses Mycobacterium tuberculosis survival in macrophages”.

For infecting cells, only one MOI is constantly used in all studies. To make sure that decrease/increase in proteins are due to infection, the authors should have used different MOI of infection at least in one place, may be at the beginning.

The expression of MFN2 was decreased in MOI dependent manner (Fig. 2E). This result suggests that reduced MFN2 is caused by Mtb. We revised the text as follows;

“In addition, we found that production of MFN2 was decreased in a dose-dependent manner during Mtb infection.”

The immunoblot results are not convincing. The B-actin levels vary from lane to lane in all blots. May be the authors should quantitate the levels of proteins with Image J or with similar software and express the reduction/increase in relation to B-actin. This is applicable to all immunoblots. Otherwise, the blot results are not valid.

We quantitated the levels of all proteins using the software Image J. These immunoblot results are convincing now. Thank you so much for your advice.

In section 3.2, the sub-title needs to be modified as "Mtb-mediated MFN2 decrease is associated". The manuscript does not discuss why the attenuated H37Ra strain is more effective in affecting the mitochondrial proteins than the virulent H37Rv strain. This is a major drawback.

We revised our subtitle with “MFN2 is sensitive to the survival of attenuated Ra.” A previous our study showed that Ra-induced ER stress is strong comparing to Rv (Lim YJ et al., Sci Rep (2016) 6:37211). The reason why attenuated strain Mtb H37Ra induces strong ER stress response is that strong ER stress affects Parkin production leading to the degradation of MFN2 through ubiquitination. We revised the Discussion.

Reviewer 3 Report

This study investigates the role of mitochondrial dynamics and ER stress on viability and bactericidal activity of murine macrophages infected with M. tuberculosis (Mtb). The authors use an in vitro model consisting of murine bone marrow-derived macrophages or RAW cells infected with the virulent H37Rv Mtb strain or its attenuated isogenic counterpart H37Ra. They found evidence of mitochondrial dysfunction in Mtb-infected macrophages, including reduced mitochondrial membrane potential, reduced oxygen consumption and increased fragmentation; this was more marked with H37Ra than H37Rv infection. They also found reduced levels of Mitofusion 2 (MFN2) which promotes mitochondrial fusion with concomitant increased levels of proteins involved in fission, including Drp1 and phospho-MFF. The decreased levels of MFN2 were associated with increased levels of Parkin which appears to ubiquitinate MFN2 leading to its degradation by the proteasome. The results of siRNA knockdown experiments and overexpression of MFN2 suggest that degradation of MFN2 during mycobacterial infection promotes mitochondrial fragmentation, macrophage apoptosis and reduces the growth of intracellular mycobacteria.

The results are interesting and the study is well performed, however, I have a few concerns/queries about the experiments. In addition, some of the results/discussion points are difficult to follow, possibly due to typographical errors.

Comments:

One of the central thesis is that mitochondrial fusion/fission is instrumental in determining the fate of both the macrophages and the intracellular mycobacteria. However, the degree of fragmentation has not been quantified. Unless this is quantified in some way, it is not valid to say that there are differences between H37Rv and H37Ra. Also, please clarify whether Ra or Rv caused more fragmentation (see lines 182/183 and 288/289). There doesn’t seem to be any difference in growth of Ra and Rv in BMDM. Were they compared side by side? Would you not expect to see differences in intracellular growth between the strains if mitochondrial function is a determining factor in this phenotype?

Reduced oxygen consumption and mitochondrial ATP production could be due to metabolic reprogramming resulting in a switch to glycolysis to produce energy and maintain cell viability. This should be mentioned in the manuscript. Alternatively, since H37Ra causes more apoptosis than H37Rv, is the greater reduction in oxygen consumption caused by Ra compared to Rv due to there being more cell death with this strain?

The western blots shown in Fig 2B and C do not really support the notion that the level of MFN2 was decreased at 48 h of infection (Line 184/185). The levels of MFN2 protein are markedly increased at 48 hours compared to 24 hours in the uninfected BMDM (as well as H37Rv-infected) – why is this? Was there some artefact of tissue culture causing this increase? It does not seem to be the case in the blot shown in Fig 3. In addition, the levels of DRP1 appear to be lower in H37Rv-infected BMDM even though it is stated that it is increased (Lines 186/187). The differences in band intensity are minimal though, so quantification by densitometry is needed to draw meaningful conclusions from these data.

Fig 3 C: Quantification of MFN2 staining intensity is needed to draw any conclusions as to its expression levels from confocal images.

How was the MOI determined? Does it refer to the ratio of bacteria:macrophages? This should be explained in the methods. What was the level of infection obtained after 3 hours incubation at this MOI?

How was apoptosis determined? Usually the AnnexinV+/PI – cells are considered to be apoptotic, is that the case here? Please explain in the methods and show some flow cytometry raw data i.e. dot plots for Annexin vs PI.

Lines 143-145: How were the concentrations of the inhibitors arrived at for use in the metabolic flux analysis? Was there any titration done to determine the optimal concentrations for these cells?

AEBSF is not a very specific inhibitor of ATF6 as it inhibits serine protease activity and may influence many other processes other than ATF6 activity. The authors should consider using a more specific inhibitor if one is available.

Do the authors think that their findings will be applicable to human macrophages? This should certainly be checked in the future, if not for this manuscript.

Figure 1E – fix the labels on the X-axis.

Author Response

Reviewer 3

One of the central thesis is that mitochondrial fusion/fission is instrumental in determining the fate of both the macrophages and the intracellular mycobacteria. However, the degree of fragmentation has not been quantified. Unless this is quantified in some way, it is not valid to say that there are differences between H37Rv and H37Ra. Also, please clarify whether Ra or Rv caused more fragmentation (see lines 182/183 and 288/289). There doesn’t seem to be any difference in growth of Ra and Rv in BMDM. Were they compared side by side? Would you not expect to see differences in intracellular growth between the strains if mitochondrial function is a determining factor in this phenotype?

Thank you for your comment. To be clarified, we quantified mitochondrial fragmentation in figure 2B. Mitochondrial fragmentation was more increased in Ra-infection BMDMs than Rv.

We revised the results and changed the figure 6B, C, E, and F. To be clarified, intracellular survival assay was expressed as the fold change in bacterial numbers at 24 h and 48 h relative to the number of initial invasive bacteria.

Reduced oxygen consumption and mitochondrial ATP production could be due to metabolic reprogramming resulting in a switch to glycolysis to produce energy and maintain cell viability. This should be mentioned in the manuscript. Alternatively, since H37Ra causes more apoptosis than H37Rv, is the greater reduction in oxygen consumption caused by Ra compared to Rv due to there being more cell death with this strain?

Absolutely, it is very important points. We think that glycolysis might be increased in Ra-infected macrophage comparing to Rv. When we evaluated the glucose uptake ratio of macrophages infected with Rv or Ra, glucose-uptake ratio was higher in Ra-infected macrophages than Rv. We will perform further study of metabolism during Mtb infection.

Reviewer’s only)

Fig. 1. Comparative analysis of glucose uptake from macrophages infected with Rv or Ra. BMDMs were infected with Rv or Ra for 24 h or 48 h, and the cells were incubated with 2-NBDG (100 μM) at 1 h. The ratio of glucose uptake measured by flow cytometry. All data are representative of three independent experiments. Statistically significant differences are indicated as follows: *p <0.05, ** p <0.01, and ***p <0.001.

The western blots shown in Fig 2B and C do not really support the notion that the level of MFN2 was decreased at 48 h of infection (Line 184/185). The levels of MFN2 protein are markedly increased at 48 hours compared to 24 hours in the uninfected BMDM (as well as H37Rv-infected) – why is this? Was there some artefact of tissue culture causing this increase? It does not seem to be the case in the blot shown in Fig 3. In addition, the levels of DRP1 appear to be lower in H37Rv-infected BMDM even though it is stated that it is increased (Lines 186/187). The differences in band intensity are minimal though, so quantification by densitometry is needed to draw meaningful conclusions from these data.

We revised the figures with good quality in figure 2. As result of repeated experiments, we found that MFN2 was decreased in 48 hours during Rv infection. Also, we found that DRP1 was increased in Ra-infected macrophages but not Rv-infected cells compared to the control.

Fig 3 C: Quantification of MFN2 staining intensity is needed to draw any conclusions as to its expression levels from confocal images.

We inserted quantification data of MFN2 expression in figure 3C.

How was the MOI determined? Does it refer to the ratio of bacteria:macrophages? This should be explained in the methods. What was the level of infection obtained after 3 hours incubation at this MOI?

Previously, we found that induction of ER stress and apoptosis were increased in Mtb-infected macrophages (PLoS One. 2011;6(12):e28531. doi: 10.1371/journal.pone.0028531. Epub 2011 Dec 14.). Thus, we performed most experiments at an MOI of 1. Multiplicity of infection (MOI) is a frequently used term in bacteriology which refers to the number of bacteria that are added per cell during infection. Phagocytosis rate was about 20 % in Rv- or Ra-infected BMDMs (Supplementary figure 1.)

How was apoptosis determined? Usually the AnnexinV+/PI – cells are considered to be apoptotic, is that the case here? Please explain in the methods and show some flow cytometry raw data i.e. dot plots for Annexin vs PI.

We calculated the sum of AnnexinV+/PI- and AnnexinV+/PI+ cells population as a apoptosis group. To be clarify, we revised supplementary figure 3 with the AnnexinV/PI staining results (flow cytometry raw data).

Lines 143-145: How were the concentrations of the inhibitors arrived at for use in the metabolic flux analysis? Was there any titration done to determine the optimal concentrations for these cells?

Our experiment was performed according to the Agilent’s guideline for using XF24 Extracellular Flux Analyzer and XF24 Cell Culture Microplates, and a previous report (Immunity. 2015 Jul 21;43(1):80-91. doi: 10.1016/j.immuni.2015.07.003.). Titration study was performed for determination of optimal concentrations of the inhibitors.

AEBSF is not a very specific inhibitor of ATF6 as it inhibits serine protease activity and may influence many other processes other than ATF6 activity. The authors should consider using a more specific inhibitor if one is available.

Generally, AEBSF has been used as an ATF6 inhibitor in many studies. We could not find some specific inhibitor of ATF6 no yet. Maybe, next time we can try it.

Do the authors think that their findings will be applicable to human macrophages? This should certainly be checked in the future, if not for this manuscript.

We strongly agree with you. We have a plan to get IRB before starting our work. Next experiments should be performed with human macrophages. Thank you so much.

Figure 1E – fix the labels on the X-axis.

Thank you so much! We revised it in Figure 1.

Round 2

Reviewer 1 Report

The paper has been revised and improved in response to my comments.

Reviewer 3 Report

Please specify in the Methods or legend to Figure S3 whether the data in Fig 6 A and D represents early apoptotic cells (i.e. Annexin V positive/ PI negative).